# Hearing Their Voices: Self Advocacy Strategies for People with Intellectual Disabilities in South Africa

**Cole Goldberg *** and **Sharon Kleintjes**

Department of Psychiatry and Mental Health, University of Cape Town, Cape Town 7700, South Africa
* Correspondence: colegoldberg.ot@gmail.com

**Abstract:** This study investigated strategies for people with intellectual disabilities to self-advocate for inclusion of their priorities in social policy processes in South Africa. *Method:* Self advocacy strategies were identified through a scoping literature review, a review of self advocacy toolkits and semi structured interviews with people with intellectual disabilities and other stakeholders working at non-governmental and disabled people's organisations. These data sources were triangulated to identify strategies to upskill and support young adults with intellectual disabilities to share their opinions and perspectives to deepen the diversity of voices engaged in social policy advocacy. *Results:* Data triangulation identified three core strategies for self advocacy, in person, written strategies and engagement through social media. *Discussion:* Inclusion of people with intellectual disabilities in civic and political life is crucial, and will only be achieved if self advocates are accepted into the policy-making arena. The cycle of perpetuating exclusion needs to be disrupted, to give people with an intellectual disabilities a say in policy decisions that have an impact on their lives. *Conclusion:* Adopting strategies which enable the inclusion of the voices of people with intellectual disabilities in civic activities holds potential for diversifying perspectives brought to public participation in policy development and implementation, which is currently primarily the domain of non-disabled citizens.

**Keywords:** intellectual disability; self advocacy; toolkit

## 1. Introduction

Persons with intellectual disabilities comprise an estimated 1–2% of the world's population, most of whom live in low-and middle-income countries (LMICs) [1]. Adnams (2010) notes that in the South African context, where this study took place, there are a lack of reliable data on the prevalence of intellectual disabilities due to exclusion in local epidemiological studies, and inadequate inclusion in census and routine data collection for disabilities in the country [2]. Adnams (2010) estimated that the prevalence rate of intellectual disabilities in South Africa may be higher than in other LMICs due to high rates of preventable causative conditions, such as nutritional deficiencies, tuberculosis meningitis, foetal alcohol spectrum disorder, violence and trauma [2].

South Africa is considered an upper middle income economy, yet it is regarded one of the most unequal countries in the world, with significant disparities between a small affluent segment of the population and the large majority of citizens, the largest proportion of whom are unemployed, and living below the poverty line [3]. According to Statistics South Africa (2019), approximately two-thirds of the population of South Africa, including people with disabilities, are reliant on their own meagre resources or state support for their basic health and wellbeing needs [4]. Yet, despite a robust policy foundation for disability [5–7], the needs of people with disabilities, especially intellectual disability, for health, social service, education and other support, remain a low public service priority in terms of implementation and resourcing of these policies. [7–10].

The Western Cape Forum for Intellectual Disability [11] is a relatively small non-governmental sector working with people with intellectual disabilities in South Africa.

They have engaged in concerted action to address the state's constitutional mandate to address the needs of children and adults with intellectual disabilities, primarily with service providers and families driving these initiatives. Adults with intellectual disabilities are increasingly being encouraged to take more control of their lives through a range of goal-setting, choice- and decision-making opportunities [12]. This shift encourages "advocacy with" rather than "advocacy on behalf of", which is still in early development in South Africa, primarily through the efforts of nongovernmental organisations (NGOs) supporting people with intellectual disabilities [13].

There are many definitions of the term "self advocacy", but Tilley, Strnadová, Danker, Walmsley and Loblinzk, (2020, p. 1152) found that the most common components include "the notion of speaking up for yourself or others, standing up for your rights, making choices, being independent and taking responsibility" [14]. The core principle of self advocacy is that all people have the right to make decisions and choices, to stand up for themselves to improve their quality of life [13].

Self advocacy efforts can take place within every sphere of a person's life, from the more immediate and personal sphere, through to the wider- ranging socio- political sphere [3]. While advocating for oneself can occur in every life situation, at home, at school and within one's local community, participation at the societal, in particular the political level, is less common for people with intellectual disabilities, particularly as their role as citizen seems to still be quite newly acknowledged [15]. There are significant barriers to participation in social policy for people with intellectual disabilities, who often experience attitudinal barriers, physical, structural and procedural barriers to participation [14,16].

People with intellectual disabilities must deal with systemic barriers which limit opportunities for participation in political life, being excluded from voting, limited social and education opportunities to prepare for this level of participation. Attitudinal barriers include negative stereotypes, being deemed incapable of decision making, and corresponding feelings that they are not considered a priority in the political space. This exclusion is not only found in mainstream society, but also within the well developed disabilities sector in South Africa, where the benefits of inclusion in the disabilities machinery's advocacy work does not currently extend to meaningful inclusion of people with intellectual disabilities. Physical barriers include inaccessible public spaces, particularly for persons with multiple disabilities, and lack of reliable and affordable public transport, as well as infrastructure and accommodations, for example the need for space for participation with a full time supporter. Where access is gained, lack of procedural accommodations to enhance their input to these processes can also stymie participation, for example, lack of access to materials ahead of participation opportunities to prepare presentations with assistance from supporters, the fast pace of proceedings at meetings, and the extensive use of written communication methods used at these engagements [1,14].

Personal barriers may also exclude individuals from policy level participation, with advocacy on their behalf being more appropriate and necessary, for example, where significant or multiple disabilities preclude conceptual understanding of issues beyond the persons immediate environment. Further, while a person might have the capacity for participation of this nature, it is possible that not everyone would choose to self represent. It is noted that while recent policy initiatives have moved toward empowering citizens with intellectual disabilities by recognising ability over disability, one needs to be conscious that some people with intellectual disabilities may have participation limitations due to limitations in their adaptive functioning which will not be remediated by support to participate at the political level [17]. It was noted that few self-advocates have severe and profound learning disabilities, in fact this population is largely absent in the literature on self advocacy [14], which focuses more on support and advocacy by others on their behalf. This study focuses on self advocacy by—not advocacy for—people with intellectual disabilities.

Where capacity permits, with support and accommodation, given that self-representation is central to rights advocacy, and that significant barriers to self-representation exist for those with potential and interest in participating in policy level self advocacy, efforts are needed to

overcome these barriers. Direct participation can help policy-makers more accurately reflect the diverse needs of citizens in the policies that drive investment and resource allocation.

The overall aim of this particular study was to investigate what strategies are best for people with intellectual disabilities to self-advocate for inclusion of their priorities in public social policy development and implementation in South Africa.

The research question for this study is: What practical strategies are best for people with intellectual disabilities to engage in self advocacy initiatives to influence social policy development and implementation which have an impact on their lives in the South African context?

## 2. Methodology

Data on viable strategies for people with intellectual disabilities to self advocate were collected through three qualitative methods:

(a) A scoping literature review to identify local and international strategies for the self advocacy for people with intellectual disabilities;

(b) A grey literature review of existing South African and international NGOs- and disabled people's organisations (DPOs) based toolkits on strategies for people with intellectual disabilities to self advocate;

(c) Semi-structured interviews to explore the perspectives of professionals at NGOs and DPOs internationally and locally, and of people with intellectual disabilities on their experience of self-advocacy.

### 2.1. The Scoping Review

The review question was: What is known about self-advocacy strategies for people with intellectual disabilities around social and health related public policy development and implementation? The review was conducted using the five steps for scoping reviews suggested by Arksey and O'Malley [18], these steps are: (1) identifying the research question, (2) identifying relevant studies, (3) study selection, (4) charting the data, and (5) collating, summarising and reporting the results. Inclusion criteria were qualitative and quantitative studies on self advocacy by people with intellectual disabilities in English across lower, middle- and high-income countries published in peer-reviewed academic journals from 2010 to 2020. Exclusion criteria were studies which focused on people without intellectual disabilities, children and adolescents with intellectual disability, non-English articles, and articles that did not focus on self advocacy strategies. Search terms employed were: (a) intellectual disability or intellectual disabilities or mental retardation or learning disability or learning disabilities or developmental disability or developmental disabilities; (b) AND strategies or methods or techniques or interventions or best practices or tool or toolkit; (c) AND self advocacy or self advocacy capacity; (d) AND policy or policies or law or laws or legislation. Databases searched were PUBMED, Scopus and Ebscohost with Academic Search Premier, Africa-Wide, CINAHL, MEDLINE, CINAL complete and MEDLINE complete searched in Ebscohost. In addition, ERIC and the Web of Science were searched for any other relevant articles. Sourced articles were reviewed by co-reviewers by title and abstract ($n = 40$) then included abstracts ($n = 15$) by full text using in-and exclusion criteria to derive the final list for inclusion ($n = 7$).

### 2.2. Review of Existing Toolkits

There are many characteristics of grey literature searches that make it difficult to search systematically particularly as there is an absence of a 'gold standard' and few other resources on how to conduct this type of search [19]. This step of the study involved a focused internet-based search for existing toolkits that address self advocacy for people with intellectual disabilities. The conduct of the review was adapted using the five steps for scoping reviews suggested by Arksey and O'Malley (2005) as noted previously [18]. The review question was: What strategies are suggested in available toolkits to support self advocacy by people with intellectual disabilities?

The search terms used were: self advocacy AND toolkit AND intellectual disability.

A toolkit was included if it was a manual or toolkit, was about people with intellectual disabilities and identified strategies for self advocacy. Toolkits were excluded if they were not about people with intellectual disabilities or were websites and not toolkits or manuals. As Godin et al. (2015) explain [19], abstracts are often unavailable when searching grey literature, and in this case an executive summary or table of contents was screened, then read full text in order to more clearly determine relevance for inclusion. This resulted in 9 included toolkits for review. An initial hand coding was necessary to develop and refine the coding framework for the qualitative analysis. The framework designed by Test, Fowler, Wood, Brewer and Eddy (2005) was used to guide the analysis and form an initial coding frame [20]. While this formed the foundational frame, additional themes were also developed throughout the process.

### 2.3. Interviews

**Development of the interview schedule:** The interviews aimed to explore the key perspectives which impact self advocacy for people with intellectual disabilities. The questions were derived from the reviews above. Two pilot interviews were conducted with key informants to assess time taken for the interview, whether the interview schedule was appropriate and understood, and whether the questions adequately elicited data, which informed the research question.

**Recruitment of participants:** Purposive, criterion sampling was used, as each participant needed to fulfil certain criteria, either that they were a person with intellectual disabilities who self-advocates on behalf of themselves and others, or professionals or trainers or supporters or organizational workers involved in self advocacy training of people with intellectual disabilities. Inclusion criteria for participants with and without intellectual disabilities were that they were over the age of majority in their country, able to understand the study aims and to provide informed consent, able to communicate verbally in English, and had experience of self advocating. The inclusion criteria excluded people with intellectual disabilities who would need someone to advocate on their behalf rather than support them to advocate for themselves, as the focus of the study is self advocacy, not advocacy. Participants with intellectual disabilities recruited for the study, were therefore all people who would be able to advocate for themselves.

Due to COVID-19 restrictions on contact and movement, two recruitment strategies were used:

(a) The owners and writers of self advocacy toolkits and manuals at organisations identified during the toolkit review were contacted via email to ascertain their interest in and to obtain informed consent for their participation in interviews.

(b) The researcher identified Facebook groups which were set up for Self-Advocates with Intellectual Disability and their supporters. The nature and aim of the study was posted and interested community members invited to contact the researcher for more information. This process led to recruitment of 9 key informants with and without intellectual disabilities.

Table 1 describes the interview participants.

**Data collection:** Commonly in qualitative research, an emphasis is placed on data saturation, and interviewing continued until no new information was gleaned [8]. Interviews took place both in person and via online platforms.

**Data analysis:** Braun and Clarke (2006)'s six-phase guide on the conduct of a thematic analysis was used [21]. These steps include: familiarization with data, generating initial codes, searching for themes, reviewing themes and codes, and write up.

Once the pilot interviews were conducted, transcribed and transcriptions reviewed for accuracy by the primary researcher (CG), CG, along with a research assistant independently read through the 2 transcripts several times to obtain an overall view of what themes may be emerging from the data and coded these interviews using Atlas-TI software (version 9.1.1 (2022), Scientific Software Development GmbH, Berlin, Germany, accessed

via the university) [22]. Open axial coding was conducted, and common ideas and themes were documented and then reviewed jointly to arrive at one consensus-based coding frame. This initial coding frame was used for further thematic analysis by C.G. Remaining transcripts were again read through several times by the researcher to obtain an overall view of what themes may be emerging from the data, coding these into the initial coding frame, and elaborating and reorganising codes where similar themes and new subthemes were identified.

**Table 1.** Study Participants.

| No. | Pseudonym | Brief Description |
|---|---|---|
| 1 | Meredith & Mark | Meredith and Mark are community health workers in an organisation in Europe internationally. They have founded a self-advocacy programme and have extensive experience of self advocacy work with people with intellectual disabilities. They had recently developed and piloted a self-advocacy toolkit to support their work at the time of the Interview and were eager to train people to run the programme internationally. They were contacted via email and replied that they were interested in being interviewed. |
| 2 | Lexi | Lexi is a self advocate with cerebral palsy and an intellectual disability who replied to the Facebook advert. She was eager to be a part of this project and have her voice and opinion heard as a disability service user. She has experience as a radio show host and self-advocating in the public sector in America. |
| 3 | Zola | Zola is a community health worker and therapist at a local South African non-governmental organisation. She has experience working with people with intellectual disabilities and experience working with self-advocacy groups for people with intellectual and psychosocial disability. |
| 4 | Ben | Ben is a young man with an intellectual disability who is living and working in the community in South Africa. He has recently completed a learnership programme which included self-advocacy training. He opted to participate to discuss his self-advocacy efforts during a recent hospital admission and his experience at his local police station. |
| 5 | Richard | Richard is a manager of a South African NGO which offers training programmes and learnerships for people with intellectual disabilities. These programmes are registered with the local education department, so this ensures that learners obtain a qualification. As part of their curriculum, the organisation tackle issues of advocacy and self-advocacy. Richard has co-facilitated advocacy events for people with intellectual disability. |
| 6 | Alex | Alex is a young man with an intellectual disability who is living and working in South Africa. He has recently completed a learnership programme which included self-advocacy training. He opted to participate to discuss his self-advocacy efforts and his experiences as a person with a disability in his community context. |
| 7 | Ellis & George | George is a young man with Down's Syndrome, and an intellectual disability from America who has become an international self-advocate on social media. He has become an internet sensation and posts regularly. His mother, Ellis, was present as a supporter during the session. |
| 8 | Derek | Derek was reached through snowball sampling, as a person that one of the other participants had worked with previously. He has an intellectual disability and is currently working at a local organisation in South Africa and acts as the self-advocacy representative for his organisation. |
| 9 | Izzy | Izzy was also reached through snowball sampling, as a person with whom one of the other participants had worked. She has an intellectual disability and is currently working at a local organisation in South Africa. She is an active self advocate. |

## 3. Results

The three sources of information were triangulated. The following image was developed to depict the themes which emerged across the data sources.

It is evident in the Figure 1, that the strategies that were suggested in the literature reviews were similar to that of the interview participants, with the interviewees suggesting two further strategies. The interviews elicited more information about the barriers and enablers to participation, as well as more detail around strategies that are best for self advocacy for people with intellectual disabilities.

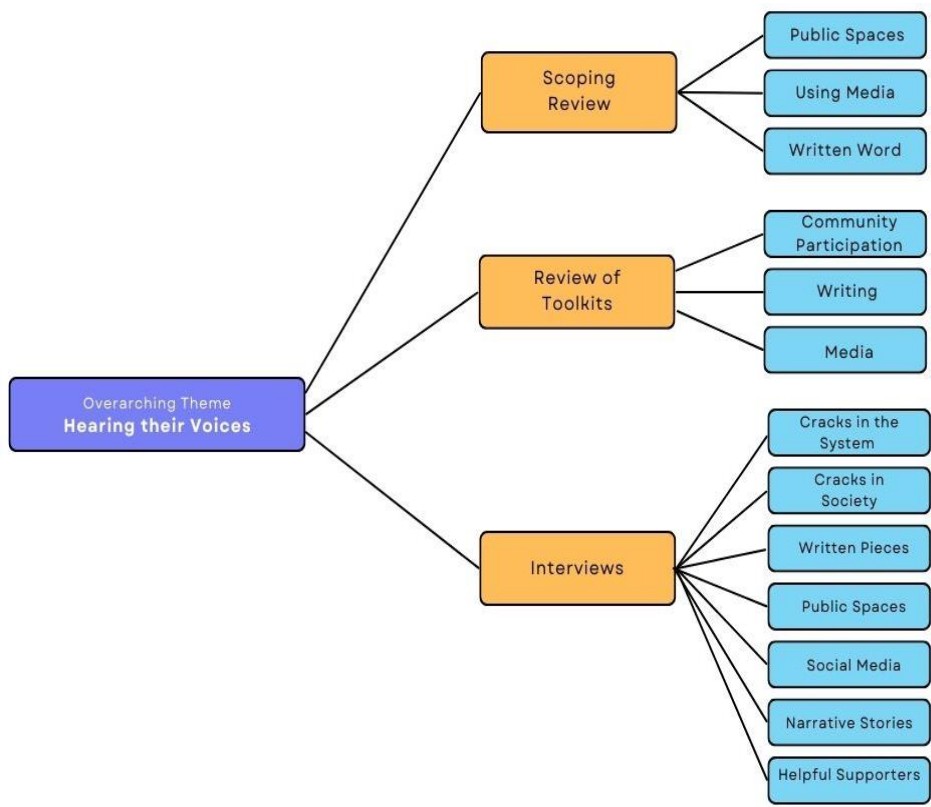

**Figure 1.** Triangulation of themes.

The overall theme entitled "Hearing the Voices" centred around the need to hear the voices of people with intellectual disabilities. This overall theme highlighted the need to no longer silence disabled voices and to provide spaces for peoples' voices to be heard. The following quotes from Richard and Izzy demonstrate the historical silencing of voices of persons with intellectual disabilities and the growing pushback by self-advocates challenging these exclusionary views:

> *"you know, it starts off with the understanding that people with intellectual disabilities have a voice . . . . why would we exclude people with intellectual disabilities from the conversation?"*
>
> —*Richard*

> *You know, this time in the world, there are people who don't regard people with disabilities as advocates or don't seem as an advocate to them. So, they can actually tell them, no, I am a person and I can, I can talk for myself. No one can make decisions for me, I can make my own decisions*
>
> —*Izzy*

The first subtheme in the interviews was entitled "*Broken Cracks*" which addresses the limitations that are placed on participants being able to self advocate, at organisational ("Cracks in the System") and societal levels ("Cracks in Society") as evidenced in Figure 1.

Participants in this study repeatedly noted that they continue to feel excluded from decision-making which has an impact on their lives. They, and other people with intellectual disabilities, still live in a society in which prejudice and stigma continue to have an impact on their exclusion, including the exclusion of their self-identified priorities for public social policy directions. Richard echoed this sentiment, and highlighted the lack of integration despite transformation, in that people with intellectual disabilities are often present in the communities but are not participating in society. He spoke about how the legislation is present, but is not being actualised.

*"I've said this before on various occasions we've reached transformation but we haven't reached integration so transformation means we've got the legislation that says certain things with regards to discrimination, specifically discrimination in the workplace and those sort of things which is which is what we deal with (at the NGO) but we haven't been able to reach integration's as yet so we still struggling with the part where we go we have this legislation which is the transformative part of it but we haven't integrated that into our mainstream society"*

*—Richard*

Ben spoke about stigma and how this impacts his identity as a person with an intellectual disability and how people with intellectual disabilities are often infantilised and are perceived as less than other people.

*Ben: It's views that are given by other people, that make people think like that ... that intellectual disability is not right and stuff like that, (later) ... You are baby. You are a baby ... you act like a baby and stuff like that.*

These Broken Cracks were also reflected indepth in both the literature reviews. International initiatives, including policy directives to address the lack of social and political participation for people with intellectual disabilities have had limited success [14]. In the interview with Zola, she reflected on the lack of support for people with intellectual disabilities in our South African context. In the literature reviews, this theme was present where the 7 included articles in this review were all from high income countries, four from Australia, one from the United Kingdom (UK), one from Canada, one from the United States of America and one multi-country study (including Ireland, Australia and the UK). In the review of toolkits, eight out of the nine toolkits were developed in two high income countries, namely the United States and the United Kingdom. Only one toolkit was developed in middle income country, Uganda, with this toolkit also implemented in a pilot project in South Africa.

In this interview Zola acknowledges the number of challenges South Africa faces, where people with intellectual disabilities are not seen as a priority to support.

*"In our country we are constantly making excuses about the high unemployment and how people with disabilities are at the bottom of the food chain"*

*—Zola*

The second subtheme of the interviews was names: "*Powerful Tools*" details participants' strategies associated with self advocacy and how they can be actualized. This subtheme was built upon the following categories identified in Figure 1: "Written Pieces", "Public Spaces", "Social Media", "Narrative Stories", "Helpful Supporters".

Writing strategies were highlighted as a viable strategy by several of the participants.

There were a number of writing strategies identified, including letter writing, position pieces, and documents, but the underlying theme was that all participants placed value on the power of the written word. Lexi spoke about her experience as an author and radio show personality. She writes articles, and position pieces on her experience of being a person with a disabilities. She also spoke of other concerns which were important for her to write and advocate about, beyond being a person with intellectual disabilities, highlighting the need to see people with intellectual disabilities as the complex multidimensional people they are, rather than narrowly framing their lives and interests only through the lens of their intellectual disabilities. Using the written word was a significant theme, in both the scoping review and the review of toolkits as well and this referred to writing opinion or position letters, drafting policy documents, writing letters of complaint, or contributing to or writing position papers.

Where participants struggled to read they noted the importance of supporters to assist them. Izzy described how her supporter would help her compile her presentations and assist her with the reporting.

Strategies for self advocacy engagement in public spaces were described by several participants. Ben highlighted the opportunities to access public spaces and be present in the communities by suggesting strategies such as protests or demonstrations. Lexi described being present on various panels and being present and visible in several interviews. Richard described an example where his institution had experience in arranging a placard demonstration on gender based violence initiated by young adults with intellectual disabilities and how valuable it is to be seen.

*"Now for me, when it was when I actually had to do a presentation or speech or talk for myself, then I at first, my first time I was nervous. Because I'm standing in front of, how can I say, 100 people then I have to give my life story. However, then later on again I got used to it because yeah, because it's people that I know will be in the audience and they now want to talk to me"*

—*Izzy*

Meredith explains how she would create opportunities for self advocates in the training programme to engage in the community. She tells a story of one particular group participant:

*"And you know he would just love having the job of going out and I think there was something more than being located in a public space which was really important. And for some and for him to just, even you know, maybe I misinterpret, but I think for him just even just seeing the visibility of being in the public space and sometimes I would purposely not get enough milk, so he could just enjoy that experience of walking through a public cafe and feeling a sense of purpose in a regular public space."*

—*Meredith*

The importance of visibility in the community and in public spaces was echoed in the findings of the literature reviews. In the scoping review—with regard to these public spaces, Petri et al. (2020) described strategies such as organising rallies to advocate for a specific cause [23]. Frawley and Bigby (2015) spoke about hosting a concert to raise awareness; in their case it was titled "Rock for Rights". Other strategies that were identified include being able to participate in campaigns, conferences and awareness raising activities [16,23,24]. Petri et al. (2020) further describe how this can take many forms such as direct teaching or giving speeches [23]. In the review of toolkits, for in person strategies in public spaces, several toolkits suggested ways in which to increase visibility in the community, such as hosting rallies and demonstrations or non violent protests (Self Advocacy Reources and Technical Assistance Centre, date unknown) [25], and having meetings with relevant stakeholders (Advocacy Focus, 2005) [26].

Several of the participants referred to using social media as a platform to spread awareness. George and his mother named specific social media sites that have been utilised to share their perspectives and hear their voices.

*Mom: we started getting phone calls from people all over wanting to interview him, and wanting to know about him and it was just truly an overnight sensation, I guess.*

*Interviewer: An overnight celebrity.*

*Mom: Yes, 15 s of fame.*

*George: Yes, that's right . . . fame!*

The 'media' (several types of media) were also suggested as a strategy to self-advocate in all articles of the scoping review (e.g., [23,27]). Petri et al. (2020) further suggest that people with intellectual disabilities can participate in drafting laws and policies, go to ministries and city councils or can hand out written materials to the community or wider public [23]. The toolkits encouraged the use of social media platforms to convey key messages, with suggested ideas including using videos, pictures, audios and other content, on specific social media platforms. Having the presence of people with intellectual disabilities on social media will allow for these (usually silenced) voices to tell their stories.

Lexi alluded to the same point, and addressed lack of emphasis on disabled voices throughout history, and how these voices are often silenced. She provides a concrete suggestion to promote inclusion, and calls for mainstreaming of disabled voices in society. Ben reflected on the South African context, the oppression that he faces as a person with an intellectual disability and suggested that story telling could be a strategy for raising awareness.

*We need to work on the treatment of people with disabilities. Like the president said so . . . yes we do and maybe as you say to tell those stories so that we can highlight the injustice and the oppression injustice ja . . . and prejudice. With the stereotypes and stuff like that.*

Regarding the subtheme of Helpful Supporters, the role of supporters was identified by all three data sources as essential to the self advocacy movement. In the interviews, Derek spoke about the role of using supporters and how they may be necessary in scaffolding the process and helping the person with intellectual disability achieve their goals.

*Interviewer: Okay, and the other people who are part of the advocacy group, who are they?*

*Derek: They are also, a few of them is our staff members. Like the job coaches are staff members and some of them is the trainees from each workshop and the job coach leads the meeting and we have one of our general managers to sit in the meeting also.*

Participants in the study acknowledged how the supporting persons (parents, professionals, colleagues, peers) had a variety of roles, including driver, reading support, companion, editor and more. This finding was echoed in the literature reviews. Izzy furthered this point, and described her first hand experience of having a supporter who assisted her with self advocacy initiatives.

*Interviewer: And tell me, tell me a little bit about the, the, idea of the supporters. Who was your supporter? What did they do for you?*

*Izzy: Now, you see, I actually paid the supporter. Now, yes, so the supporter that would now help me with getting my reports ready, that time she was the general manager.*

*Interviewer: Okay . . .*

*Izzy: Oh yeah, so whatever she did was she, compiled my reports with me, but in a professional way on her laptop. Yeah, and then she will now sit with me and ask me, now what do I want to say? And then, (name) was my other supporter, she was actually technically in the board meeting, so she was my supporter and she was my chauffeur.*

All of the strategies suggested above highlight the importance of having the voices of people with intellectual disabilities heard, valued and respected.

*Lexi: "but if we just take the powerful tool of our ears and our voices and listen maybe we could combine those stories and make a difference"*

## 4. Discussion

Inclusion of people with intellectual disabilities in civic and political life will only be achieved if those who are willing to participate are accepted into the policy-making arena [28]. Participants in this study indicated a clear desire for the voices of people with intellectual disabilities to be heard. Self representation by self-advocates not only asserts their right to participation but can challenge negative stereotypes about people with intellectual disabilities, contributing to changing societal perceptions about the passivity and incapacity of people with intellectual disabilities [16].

Several articles suggest that with political inclusion and citizenship for people with intellectual disabilities as the goal, guiding structures, frameworks and policies are in place to promote political participation by people with intellectual disabilities, however these have not sufficiently been actualized [12,15,23,27]. This may be due to ongoing stigma and prejudice and lack of exposure to positive role models with intellectual disabilities in these spaces. Concern is often expressed about their ability to participate in these

processes [28], resulting in a lack of consultation or low influence of their views on public policy engagement directions [16]. This impacts the degree to which implementation of these social policies can reflect the diversity of needs within society. In turn, this makes it essential for policy development teams to include self-advocates with intellectual disabilities, and to provide reasonable accommodations which enable inclusion in policy related processes [29].

An essential way in which to promote the inclusion of their voices is to improve opportunities for people with intellectual disabilities to self-advocate for their priorities [30]. Lack of exposure to appropriate mechanisms for engagement in policy-making environments make it difficult for people with intellectual disabilities to participate where opportunities arise for providing input to these processes. While a supportive environment is essential for participation, it must be acknowledged that some people with intellectual disabilities' personal incapacities may preclude them from advocating at the level of policy participation [17], however, many people with intellectual disabilities can, with skills training, exposure and support, participate in these processes with reasonable accommodations, providing the very necessary exposure to the very real contribution which people with intellectual disabilities can make in the policy space. This exposure can help reduce the false rhetoric about their incapacity for participation [28].

Where people with intellectual disabilities have potential for and interest in participating in policy processes, self advocacy skills training and support in choosing a self advocacy strategy are key to enhancing policy participation to promote the inclusion of their voices [20]. While there is a proliferation of tools for improving skills of people with intellectual disabilities to self-advocate, this study found very few focusing on skills development to build participation at the level of policy making processes. Similarly, the scoping review found few papers which focused on this topic, highlighting the significant gap in the research where only seven articles were appropriate for inclusion in the study. Where data were found, articles and toolkits provided little information on the impact of these skills development tools on enhancing actual participation in policy-making processes. Further work is clearly needed, both to elaborate on ways to support participation but also to establish the impact of skills development to participate in self advocacy initiatives on influencing policy decisions.

Strategies suggested by this study for self advocacy include the use of Public Spaces, Social Media, Written Pieces, and Narrative Stories. These strategies are similar to the findings of Schmidt, Faieta and Tanner [31] who found that strategies could include interactive multimedia education, peer led group interventions, writing interventions, workplace modifications and specific health condition related programmes. While these strategies present several promising options for improving the fledging self-advocacy movement for people with intellectual disabilities in South Africa, the findings indicate that having supporters who can help with these self advocacy initiatives is paramount to the successful of these initiatives. Tilley (2013, p.472) suggests that "for self advocacy organizations to make an impact on community issues, it may be necessary to harness and support interpersonal tendencies—to develop a more effective partnership" [32]. As this work is currently done by only a few, underfunded organisations, work is needed to elaborate the roles and support structures needed for informal, family, peer and organisational supporters to expand to meet the expressed need for supporters noted by participants in this study. Research to inform this emerging practice in the local South African context is warranted.

*Limitations of the Study*

The ten year period used for the scoping review only yielded 7 papers for inclusion, and broadening the time period for the review could have included more papers for review. In addition, only English language papers were indexed. More participants with experience advocating at a social policy level may have elicited further insights.

## 5. Conclusions

The theme of mainstreaming the voices of people with intellectual disabilities is central to the findings of this study, highlighting the need for a societal shift and practical interventions to support the inclusion and valuing of diverse voices, including those of people with intellectual disabilities. A specific focus is needed, one that shifts the narrative towards people with intellectual disabilities as active participants in society.

Adopting strategies which enable this shift holds potential for addressing the need to diversify perspectives brought to public participation in policy development and implementation in South Africa, extending the current involvement of primarily non-disabled citizens to include the perspectives of people with intellectual disabilities.

**Author Contributions:** Conceptualization, C.G. methodology, C.G., validation, C.G.; formal analysis, C.G.; investigation, C.G.; resources, C.G.; data curation, C.G.; writing—original draft preparation, C.G.; writing—C.G. and S.K.; visualization, C.G.; supervision, S.K.; project administration, C.G.; funding acquisition, CG. and S.K. All authors have read and agreed to the published version of the manuscript.

**Funding:** This research was funded by the National Research Foundation (NRF), South Africa, grant number 120822. The research was also funded by the Hendrik Vrouws Scholarship, South Africa and the Vera Grover Scholarship, South Africa.

**Institutional Review Board Statement:** The study was conducted in accordance with the Declaration of Helsinki, and approved by Human Research Ethics Committee, Faculty of Health Science, University of Cape Town, HREC approval number: 850-2020.

**Informed Consent Statement:** Informed consent was obtained from all subjects involved in the study.

**Conflicts of Interest:** The authors declare no conflict of interest.

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
