# Peer review of "Hearing Their Voices: Self Advocacy Strategies for People with Intellectual Disabilities in South Africa"

_disabilities, doi:10.3390/disabilities2040042_

Round 1
Reviewer 1 Report
This is an interesting topic to research and one that has crucial practical relevance to disability-inclusive political and civic participation. The range of methods deployed are important, although I have some issues with their framing (see below). However, my central reservation with the manuscript in its present form is that it deals with several key areas too superficially.
Literature Review:
The literature review is too short and superficial. The authors should discuss what are some of the barriers that people with intellectual disabilities face in (some of) the situations noted on line 42 (i.e. at home, at school and within one’s local community, as well as in broader social contexts) that renders self-advocacy particularly important for them. This can include discussion of why people with intellectual disabilities may be more marginalised in many contexts compared to people with other disability types, which will help justify the focus of the manuscript on people with ID specifically.
In relation, the study appears concerned specifically people with IDs participation in policy decisions. But there is no discuss of the barriers that people with ID (and people with disabilities more generally) face to political and civic participation like voting, participation in local government meetings etc
The literature review should also situate disability within the South African context- for example, how many people with intellectual disabilities are there in South Africa? Do policies and laws of South Africa guarantee the civic participation of all people with disabilities, including people with ID? (Note South Africa has also ratified the UNCRPD- relevant to discuss international commitments).
Method
Line 86- this is not a review of grey literature as the search parameters are limited to toolkits and not other types. The research question offered in line 91 is really a sub-element of the one in line 64, as, quite rightly, the authors have decided that toolkits contain useful knowledge about self-advocacy strategies. Suggest to merge these sections so that the toolkit search is a sub-element of the scoping review.
Research questions would be clearer if presented at the conclusion of the literature review.
What were the characteristics of participants? How many were people with ID and what were the key pertinent characteristics/affiliations of the non-disabled participants?
Results
Presently, key interview findings are discussed superficially and data are weakly triangulated.
First, in the interview data there appears to be one theme with two subthemes. The two subthemes have two and six categories respectively. Underpinning these are only two quotes from the interview data and little further analysis, meaning that crucial contextual data is missing. For example, “Helpful Supporters”? Who are these supporters- are they family and friends or does it refer to structural/institutional support? In relation, the quotes are not attributed to any participant, so it is not clear whether they are from people with or without ID.
Second, weak triangulation. Appreciate that the authors have a lot of ground to cover, but currently, since the methodological components are presented separately, the data do not speak to each other. For example, line 175 -180 notes the written word and social media were prominent strategies identified in the toolkit review. The category names from Line 195 suggests that participants corroborated this in the interview data, but this is never made clear. Linking up/merging these different methodological components would spotlight to the reader where the strategies identified in the background review corroborate or diverge from the suggestions and lived experiences of people with ID. Moreover, the vast majority of lit identified in scoping review is from high-income contexts and non-South African, so the key informants offer crucial perspective on what self-advocacy strategies may transfer or would need adaptation to the South African civic context. Merging the analysis sections will help highlight this.
In light of above, advise:
- Present far more data and interpretation from the key informant interviews.
- Present results from scoping review and key informants together. Since the article is on self-advocacy, it would be nice to lead with the lived experiences and ideas of the participants with ID and then contextualise their suggestions with the other forms of data
- A table of themes would aid clear presentation of interview data
Finally, the manuscript makes repeated references to a toolkit having been developed from the findings. This is important and laudable, but this toolkit is not itself a finding or object of analysis of this research. The central contribution of the analysis in the manuscript is to empirically identify strategies to develop the capacity of people with intellectual disability to self-advocate within social policy processes in South Africa. How the authors have chosen subsequently chosen to operationalize the findings is outside the scope of the manuscript and confusing for the reader as the toolkit is not referenced.
Reviewer 2 Report
Dear authors, thank you for this interesting paper. You have an important topic. However, your search of the literature was too limited hence I have given your paper an 'average' rating. Your start date for the systematic review was 2010. This meant that you missed this gem of an article, which is the most detailed empirical study I know about involvement in policy making:
Redley, M., & Weinberg, D. (2007). Learning disability and the limits of liberal citizenship: Interactional impediments to political empowerment. Sociology of Health & Illness, 29(5), 767–786. doi: 10.1111/j.1467-9566.2007.01015.x.
There are some other papers which I believe warrant discussion:
Power, A., & Bartlett, R. (2018). Self-building safe havens in a post-service landscape: How adults with learning disabilities are reclaiming the welcoming communities agenda. Social & Cultural Geography, 19(3), 336-356.
Bylov, F. (2006). Patterns of culture and power after the great release: The history of movements of subculture and empowerment amongst Danish people with learning difficulties. British Journal of Learning Disabilities, 34(3), 139-145.
Tilley, E. (2013). Management, leadership and user control in self-advocacy: An English case study. Intellectual and Developmental Disabilities, 51(6), 470-481.
Fyson, R., & Fox, L. (2014). Inclusion or outcomes? Tensions in the involvement of people with learning disabilities in strategic planning. Disability & Society, 29(2), 239-254.
I could go on. I am unsure why your approach did not yield a wider range of papers, and I was surprised that there were so few illustrations in the grey literature. As you do not list those you did find it is difficult to comment further.
Beyond these issues my sense is that your discussion is a little superficial. I certainly know parents of people with more significant intellectual and other impairments who would laugh at the idea of teaching their son or daughter to engage with policy makers and policy making. Redley and Weinberg's paper indicates that even for less severely impacted individuals, there needs to be scaffolding in place to help people to engage. The education many students receive in special schools is not designed to give them political awareness.
I would urge you to spread your reading net more widely, and to strengthen your paper with a deeper reflection on the barriers to people participating, both stemming from the intellectual disability (the medical model if you like), and from barriers in the wider world (the social model). And then to address strategies to overcome these barriers.
